# Aging and Central Auditory Disinhibition: Is It a Reflection of Homeostatic Downregulation or Metabolic Vulnerability?

**DOI:** 10.3390/brainsci9120351

**Published:** 2019-12-01

**Authors:** Baher A. Ibrahim, Daniel A. Llano

**Affiliations:** 1Department of Molecular and Integrative Physiology, University of Illinois at Urbana-Champaign, Urbana, IL 61801, USA; bibrahim@illinois.edu; 2Beckman Institute for Advanced Science and Technology, University of Illinois at Urbana-Champaign, Urbana, IL 61801, USA

**Keywords:** aging, GABA, glycine, presbycusis, auditory, tinnitus, mitochondria, metabolism

## Abstract

Aging-related changes have been identified at virtually every level of the central auditory system. One of the most common findings across these nuclei is a loss of synaptic inhibition with aging, which has been proposed to be at the heart of several aging-related changes in auditory cognition, including diminished speech perception in complex environments and the presence of tinnitus. Some authors have speculated that downregulation of synaptic inhibition is a consequence of peripheral deafferentation and therefore is a homeostatic mechanism to restore excitatory/inhibitory balance. As such, disinhibition would represent a form of maladaptive plasticity. However, clinical data suggest that deafferentation-related disinhibition tends to occur primarily in the aged brain. Therefore, aging-related disinhibition may, in part, be related to the high metabolic demands of inhibitory neurons relative to their excitatory counterparts. These findings suggest that both deafferentation-related maladaptive plastic changes and aging-related metabolic factors combine to produce changes in central auditory function. Here, we explore the arguments that downregulation of inhibition may be due to homeostatic responses to diminished afferent input vs. metabolic vulnerability of inhibitory neurons in the aged brain. Understanding the relative importance of these mechanisms will be critical for the development of treatments for the underlying causes of aging-related central disinhibition.

## 1. Introduction

Aging-related hearing loss is increasing in the population. There will be an approximately 50% increase in the U.S. population over 70 years by 2050 and approximately half of them will develop hearing loss as they age [1,2,3,4]. With aging, intelligibility of complex sounds is generally impaired beyond what can be explained with threshold or tuning shifts (reviewed in [4,5,6]), suggesting that aging-related changes occur in the central auditory system that may not be solely explained by changes in the periphery. Therefore, any comprehensive effort to understand and treat aging-related hearing loss must include an approach to ameliorate central changes that occur in the aging auditory system.

A number of aging-related changes across all levels of the central auditory system have been described. For example, aging is associated with alterations in a broad range of neurochemicals across auditory nuclei, in particular the inhibitory neurotransmitters gamma-amino-butyric acid (GABA) and glycine; changes in the ability to buffer calcium; morphological changes, such as diminishment in cellular and synaptic density in the cochlear nucleus as well as loss of auditory cortical thickness; diminishment in the fidelity of responses to temporally-patterned signals; increased spontaneous activity; diminished hemispheric lateralization of neural responses; and increases in frontal cortex activation during challenging auditory tasks in humans (see [7,8,9,10] for reviews). Since virtually all studies on aging have been conducted in either humans or animals with some degree of peripheral hearing loss, it is not clear to what extent the aging-related changes listed above are driven by a loss of peripheral acoustic input compared to intrinsic aging-related mechanisms. Answering this question is critical because if the central changes are a consequence of peripheral hearing loss, then aggressive measures to rehabilitate the peripheral acoustic apparatus (e.g., early use of hearing aids or cochlear implants and/or aggressive auditory rehabilitation) would be indicated to prevent later central changes. If intrinsic metabolic changes related to cellular aging were the main driving factor responsible for central changes, then early aggressive metabolic interventions (e.g., exercise, antioxidants, more stringent treatment of metabolic syndrome, etc.) would be indicated.

Decreases in synaptic inhibition in the aging central auditory system have been interpreted to be a consequence of a loss of peripheral input and therefore to be an adaptive response to boost failing peripheral acoustic signals [11]. In this context, diminished synaptic inhibition has been interpreted as being responsible for other pathological changes, such as increased spontaneous activity, diminished temporal fidelity, and perceptual changes, such as tinnitus and diminished ability to suppress distractors [12,13]. This view comports with a growing body of literature demonstrating that neural networks homeostatically adjust their overall magnitude of excitation and inhibition to maintain a balance between the two (reviewed in [14]). This type of mechanism may elevate spontaneous firing rates in cochlear nucleus neurons, as shown via a series of computational modeling studies by Schaette et al. [15,16,17]. Although a homeostatic explanation may be applicable at the level of the cochlear nucleus, which receives diminished inputs from the eighth nerve in aging-related hearing loss, it is less clear how well it applies beyond this nucleus. Given the increases in spontaneous activity and acoustic responsiveness seen in the aging cochlear nucleus compared to younger animals [18,19,20,21,22], higher centers may actually receive a compensated, or even an increased, degree of excitatory synaptic input. An alternative hypothesis is that the decreases in synaptic inhibition that are seen with aging, particularly in higher centers, are due to metabolic vulnerabilities in inhibitory neurons, particularly GABAergic neurons. As will be reviewed below, GABAergic neurons appear to have higher energy requirements than excitatory neurons, and GABAergic neurons are diminished across a host of aging-related diseases, suggesting that decreases in GABAergic function in the aging central auditory system may be, at least in part, caused by their high metabolic demands, rather than as a compensatory response to diminished sensory input. Below, we review the evidence for each of these theories, and suggest experiments to test them.

## 2. Synaptic Inhibition in the Aging Central Auditory System

Inhibition plays a key role shaping response properties in the auditory system. Synaptic inhibition has been shown to be critical for frequency tuning [23,24,25,26,27], duration tuning [28,29,30], response timing [31], sound localization [32,33], and responses to temporally patterned sounds [34,35,36,37]. Thus, loss of synaptic inhibition would be expected to interfere with perceptions of small differences in pitch, in sound localization, and with the ability to perceive rapid fluctuations in speech (which are critical for distinguishing many consonants and consonant–vowel transitions), all of which are impaired during aging [38,39,40,41,42,43]. Therefore, aging-related alterations in the magnitude of synaptic inhibitory currents in the central auditory system likely disrupt a broad array of processes necessary for experiencing the richness of the acoustic landscape.

Loss of synaptic inhibition with aging is seen from the cochlear nucleus to the auditory cortex and has been observed across rodent and primate species. The animal models chosen for this work are a significant variable to be accounted for. In mice, two strains have been used for most studies: C57BL/J and CBA/CaJ. C57BL/J mice begin to lose hearing early, at approximately six months of age, due to the presence of a mutation in the cadherin 23 gene, while CBA/CaJ mice retain relatively good hearing until later ages [44,45]. Thus, comparisons between C57BL/J mice and CBA/CaJ mice have been used to understand the relative contribution of hearing loss compared to chronological aging in the central auditory system. Similarly, Fischer 344 rats have relatively early hearing loss compared to other strains [46], and have been used in comparison with other strains to deduce the contribution of hearing vs. chronological factors, as outlined below.

Experimental determination of the relative contribution of hearing loss vs. chronological age to changes in inhibition is not straightforward. At a minimum, one would expect to see a correlation between hearing sensitivity and a measurement of synaptic inhibition (e.g., GABA synthetic enzyme expression, receptor binding, etc.). However, since age and hearing loss are correlated in most species, the hearing loss/synaptic inhibition relationship is confounded by age. More appropriate approaches would be to conduct a multiple regression of these factors to estimate their individual contributions, or to induce hearing loss in young animals, removing the variable of aging. Few studies have used these approaches, and have led to conflicting results, reviewed below. Further, it is important to note that induced hearing loss, typically done via noise exposure, leads to an increase in physiological and psychological stress, which may create a neuroendocrine environment that causes dysfunction selectively in metabolically vulnerable cell types, such as GABAergic neurons. Thus, a simple demonstration of loss of inhibition with aging in a species that shows aging-related hearing loss does not lead to a straightforward understanding of the factors causing the decrease in inhibition.

### 2.1. Auditory Brainstem Changes

The cochlear nucleus, which is the first station in the central nervous system to receive information from the auditory nerve, and has both glycinergic and GABAergic inhibitory neurons, has been shown to have an aging-associated decrease in glycine binding sites [20,21] and decreased expression of glycine receptor subunits [21,22]. These decreases were seen in Fischer 344 rats older than 18 months, which show early hearing loss with aging [46], though in these studies, correlations between hearing loss in individual animals and glycine receptor expression or binding were not measured. In addition, glutamate, glycine, and GABA levels are severely depressed in the cochlear nucleus of 33-month-old Fischer Brown Norway rats (an age and strain that shows significant hearing loss [47]), compared to younger animals [48]. Similar decreases were seen across the other central auditory nuclei (superior olive, inferior colliculus, auditory thalamus, and auditory cortex) in this study, though the changes were most pronounced in the cochlear nucleus and superior olive. To determine if hearing loss is a driving factor in the downregulation of inhibition, Whiting et al. temporarily disrupted hearing unilaterally using earplugs and examined glycine expression on the plugged and unplugged sides. They observed a drop in expression of the GlyRa1 subunit of the glycine receptor in both bushy and fusiform cells on the plugged side and found a return to baseline post-removal [49]. Other aging-related changes, such as morphological changes in the octopus cell area, showed roughly similar aging-related trends in C57BL/6J and CBA/J mice, whose hearing thresholds differ greatly [50]. These data suggest that at least at the cochlear nucleus, hearing loss itself may drive significant downregulation of glycinergic transmission, whereas chronological factors may drive morphological changes.

Beyond the cochlear nucleus, the medial nucleus of the trapezoid body, a glycinergic nucleus in the auditory brainstem, undergoes a decrease in its number of synaptic terminals and neurons in the aging (24-month-old) Sprague-Dawley rat [51,52,53], though again, correlations with hearing loss were not measured in this study. Only one study, to our knowledge, examined aging-related changes in neurotransmitter synthesis in the nucleus of the lateral lemniscus in the aged (24–36 month) Fischer 344 rat, and they found an aging-associated decrease in the activity of the main synthetic enzyme for GABA, glutamic acid decarboxylase (GAD) [54].

### 2.2. Auditory Midbrain Changes

Several studies have been done to examine aging-related changes in synaptic inhibition in the auditory midbrain. Several of these have found an aging-associated decrease in GABA synthesis (as measured by decreases in GAD expression) [55], diminished GABA levels [48,56,57], and decreased GABA receptor expression [58,59,60]. In none of these studies did the authors measure correlations between hearing loss and changes in GABA levels, synthesis, or receptor levels. To test the impact of hearing loss more specifically, Milbrandt et al. induced acoustic trauma in three-month-old Fischer 344 rats (106 dB SPL [sound pressure level] 12 kHz pure tone for 10 h), leading to primarily outer hair cell loss near the base of the cochlea. The authors observed a transient, but significant, drop in GAD levels, and a significant increase in GABA subtype A receptor (GABA_A_) receptor expression (measured by muscimol binding), as shown in Figure 1. Thus, acoustic trauma appears to cause one change seen with aging (lowered GAD levels), but the opposite of another change with aging (increased GABA receptor expression) [61], suggesting that lowered peripheral input is not solely responsible for aging-related changes in GABAergic inhibition in the inferior colliculus. Further, a more recent study compared the impact of noise to styrene-induced hearing loss on GAD expression in the inferior colliculus. Although both noise and styrene induced significant peripheral threshold shifts, only noise was associated with a drop in GAD expression in the inferior colliculus [62], suggesting that peripheral hearing alone is not responsible for inferior colliculus disinhibition. Developmental cochlear ablation also leads to diminished intrinsic inhibitory currents in inferior colliculus neurons in vitro [63], though it is not yet clear whether the same mechanisms that are engaged in developmental plasticity are also involved in aging-related changes in central inhibition.

### 2.3. Auditory Thalamus and Cortex Changes

Few studies have examined changes in thalamic inhibition with aging, but the work that has been done suggests that there is a decrease in total [48] and extrasynaptic [64] GABA with aging. In the auditory cortex, decreases in GAD65 and GAD67, two isoenzymes involved in GABA synthesis, are decreased in both aging Long-Evans and aging Fischer 344 rats, suggesting that chronological aging, rather than hearing loss, is responsible for this change [55]. The decrease in GABAergic neurons in the auditory cortex is seen in both parvalbumin-expressing and somatostatin-positive interneurons [65,66,67,68]. An examination of synaptic inhibition induced by thalamocortical transmission in a brain slice preparation revealed strong aging-related decreases in inhibition (Figure 2) but found that the strongest predictor of the decrease in inhibition was an anatomical marker of chronological age (cortical thickness) and that the addition of hearing loss in a multiple regression model had no additional ability to predict the degree of GABAergic inhibition [69]. In contrast, support for the hypothesis that hearing loss is the primary driver for decreases in inhibition is the finding that chronic (two months) exposure to noise in six-month-old Long-Evans rats induces a drop in the number of parvalbumin-positive neurons in the auditory cortex [70], in addition to work showing that juvenile-onset hearing loss diminishes the appropriate maturation of auditory cortex inhibitory neurons [71]. These data point to a mixture of hearing loss and chronological aging-related factors leading to central auditory disinhibition with aging. Below, we review energy-related factors that may increase the vulnerability of GABAergic neurons to metabolic disruptions involved with aging.

## 3. Aging and Mitochondrial Energetics

A potential connection between aging and loss of inhibitory tone is the gradual failure of cellular energy metabolism that occurs during aging. Aging mitochondria are less efficient in ATP production and in their ability to control free radicals formed during oxidative phosphorylation. Specifically, aged mitochondria have lower levels of enzymes in the tricarboxylic acid cycle, lower levels of enzymes that convert ketone bodies to acetyl coenzyme A, and diminished function of the electron transport chain [72,73,74,75,76,77,78]. In addition, the normal antioxidant mechanisms that protect tissues from the reactive oxygen species that are produced as a consequence of free radical formation are less abundant during aging [79,80]. Thus, aging is associated with progressive dysfunction of tissues, such as those found in the brain and cochlea, that have high energy demands. Accordingly, we previously found that the most potent predictors of aging-associated hearing loss are physiological metrics of the tissue redox state [81]. Specifically, we observed that the oxidized flavin adenine dinucleotide/ reduced nicotinamide adenine dinucleotide (FAD/NADH) ratio of the stratum pyramidale of the hippocampus strongly predicted hearing loss. This finding implicates deficiencies in energy metabolism as a major mediator of the cellular pathological changes seen with aging.

## 4. GABAergic Neurons are More Vulnerable to Energy Insults

All biological systems need energy for optimum function. At the cellular level, brain energy is utilized to meet the demand of certain functions, such as maintaining the resting membrane potential as well as the release and recycling of neurotransmitters [82,83]. Given that different cell types with different intrinsic properties and functions could have different energy needs, cells of high energy demand and/or lacking the defense mechanism against metabolic insults could be more vulnerable to energy deficits. Many studies have shown that most brain ATP is consumed by glutamate-mediated processes [83,84,85]. Given that synchronized and coordinated inhibition mediated by GABAergic cells can modulate the output of the principal neurons [86,87,88,89], GABAergic cells could have more energy demand than their glutamatergic counterparts. For example, parvalbumin-expressing fast-spiking GABAergic neurons were found to be the main generator of a high-frequency gamma oscillation by sending their outputs to both glutamatergic cells and other inhibitory interneurons [90,91]. Gamma oscillation in the hippocampus and the neocortex [92,93,94] has a high energy demand indicated by a positive correlation between gamma power and both glucose and oxygen utilization as well as mitochondrial complex I gene expression and high ATP production [95,96,97].

Accordingly, parvalbumin-positive fast-spiking GABAergic neurons have some unique features compared to other cells. These neurons are able to generate action potentials at frequencies >100 Hz, have reliable signal propagation and faster conductivity due to the expression of a supercritical density of Na^+^ channels in their unmyelinated axons, keep their resting membrane potential closer to spiking threshold, and have a lower input resistance and faster time constant than glutamatergic cells [91,98,99]. As reviewed by Kann et al., parvalbumin fast-spiking neurons have more stable neurotransmitter release and a low probability of conductance failure compared to glutamatergic cells as reviewed in [91]. Further, the release of GABAergic synaptic vesicles from the GABAergic synapses between dentate gyrus basket cells and principal neurons was found to have a large quantal size and a high release probability [100]. All of these synaptic mechanisms demand a significant amount of energy to maintain the ionic gradients across the cell membrane and to release neurotransmitter in a calcium-dependent manner. Consequently, inhibitory interneurons, located in the principal cell layers, stratum oriens of the CA1–3 areas, stratum lucidum, and hilus of hippocampus, have a higher number of mitochondria than the principal cells [101]. In addition, high mitochondrial density appears to be associated with high performance even within the same cell type. As such, the different types of GABAergic cells have different mitochondrial density depending on their synaptic performance. Interestingly, an examination of the ultrastructure of presynaptic mitochondria and cytochrome c in different GABAergic cell types showed that parvalbumin-positive fast-spiking GABAergic neurons have more mitochondria and a higher level of respiratory chain protein cytochrome-c compared to slow firing type-1 cannabinoid receptor-positive basket cells [102].

In addition, as the last important step of the mitochondrial oxidative chain to produce ATP [103,104], cytochrome oxidase (respiratory chain complex IV) is a good indicator of the metabolic demand [105,106]. Across different species, the colocalization between cytochrome oxidase and some GABAergic markers was reported. The metabolic hot spots in the macaque striate cortex, indicated by high levels of cytochrome oxidase, are immunopositive for both the vesicular GABA transporter and GAD enzyme [107]. Moreover, the neurons of the perigeniculate nucleus and the terminals of the basket cells of cerebellum in the cat showed a colocalization between cytochrome oxidase and GABA [108]. Also, the mouse’s inferior colliculus showed a prominent regional colocalization between cytochrome oxidase and GAD [109]. These observations suggest that GABAergic neurons are metabolically highly demanding. In addition, an increased glucose metabolism was reported upon inhibition of hippocampus pyramidal neuron firing following electrical stimulation of the hippocampal fimbria-fornix pathway [110]. The long suppression of the pyramidal cells was correlated with an increase of the hippocampal uptake of 2-deoxyglucose-(^14^C), particularly in the area full of inhibitory intraneuronal connections to the pyramidal cells [110]. Moreover, glucose metabolism was higher in GABAergic neurons, compared to glutamatergic cells, of the hamster’s somatosensory cortex in task-related activity using 2-deoxyglucose as an indicator [111].

Despite their high energy demand, GABAergic cells may lack the proper defensive mechanisms or buffering systems against energy insults, such as hypoxia or low ATP levels. An in vitro model of cerebral ischemia, made by reducing the perfusion rate of the artificial cerebrospinal fluid to the cortical slices, displayed a permanent impairment of the cortical interneurons, compared to excitatory principal cells, throughout the experiment even after reinstating the normal perfusion rate [112]. Consistent with this finding, hypoxic events caused a long-lasting reduction of the power of gamma oscillation, which is mainly generated by parvalbumin fast-spiking GABAergic cells [91,113,114,115]. Moreover, GABAergic cells could lack buffering mechanisms that result in persistent membrane depolarization under low ATP levels. For instance, cholinergic cells in the striatum have a protective mechanism that keeps them hyperpolarized by opening a calcium-dependent K^+^-channel as a response to ATP depletion. In contrast, the GABAergic spiny cells, lacking such a defensive mechanism, were kept depolarized during energy insults, which resulted in their selective loss, which may contribute to mechanisms underlying Huntington’s disease [116,117]. GABAergic cells may also be more vulnerable than excitatory cells to high levels of extracellular glutamate; a mechanism which may play a role in the progression of the neurodegenerative diseases [118]. Excitotoxicity or the persistence of receptor activation by ambient glutamate is one of the negative consequences of an energy deficit and may be observed in other pathological states, such as trauma and seizures [119,120,121]. The high levels of glutamate in the synaptic cleft leads to unregulated cellular activity and may engage the surrounding cells in vicious cycles of increasing intracellular calcium and further glutamate release [121]. Increasing the intracellular calcium levels can activate the nitric oxide synthase pathway and free radical formation [119,120]. Accordingly, the cells producing high levels of reactive oxygen species under normal conditions will be the first to be negatively affected by cytotoxicity. Having highly active mitochondria, parvalbumin fast-spiking GABAergic neurons may produce high levels of reactive oxygen species, such as superoxide and hydrogen peroxide, under physiological conditions, as reviewed in [91]. Consistent with this energy demand, oxidative stress was reported to precede the reduction of parvalbumin reactive interneurons in the hippocampus after genetically compromised glutathione synthesis [122]. Interestingly, GABAergic cells have a high expression of peroxisome proliferator-activated receptor gamma coactivator 1-alpha (PGC-1) [123], which is critical as an energy regulator and as a buffering agent of the high level of oxidative stress by inducing numerous antioxidant genes [124,125]. PGC-1α was known to induce the expression of both superoxide dismutase 2 and glutathione peroxidase 1, which remove reactive oxygen species, such as superoxide and hydrogen peroxide [126]. As such, the vulnerability of parvalbumin GABAergic cells to oxidative stress was linked with the downregulation of PGC-1 [127]. Further, GABAergic dysfunction and parvalbumin deficiency were observed in the PGC-1-knock out mouse. Consistently, many studies showed a selective reduction of GABAergic cells after excitotoxic shock or with the dysregulation of reactive oxygen species-scavenging enzymes and glutathione [128,129,130,131]. Further, GABAergic cells could be more vulnerable to mitochondrial toxins. The selective poisoning of the mitochondria of parvalbumin fast-spiking GABAergic neurons resulted in sustained membrane depolarization and a rapid decrease of gamma oscillations, with no effect on the pyramidal neurons [115]. Also, in vivo, the tricarboxylic acid cycle of GABAergic cells was selectively inhibited by its specific toxin, 3-nitropropionic acid, while those of glia and glutamatergic cells were unaffected [132].

### The Impact of the Vulnerability of GABAergic Cells on Brain Pathology

Selective vulnerability of GABAergic neurons may play a role in the development of several different pathological states. For example, preclinical animal models having DNA mutations to mimic neurological diseases, such as schizophrenia, cognitive dysfunction, and autism spectrum diseases, were characterized by increasing oxidative stress indicated by high levels of 8-hydroxy-2’-deoxyguanosine immunoreactivity, a marker of DNA oxidative damage, as well as a low percentage of parvalbumin-reactive neurons and parvalbumin-reactive cells surrounded by perineuronal nets [133]. Clinically, dysfunction of different types of GABAergic neurons has been associated with neuropsychiatric diseases, such as schizophrenia, depression, and epilepsy. For instance, some studies showed that the postmortem samples collected from patients with schizophrenia, bipolar disorder, and major depression were characterized by low levels of parvalbumin or *GAD67* gene expression [134,135], whereas some studies showed that these neurological disorders were also associated with the downregulation of somatostatin- and parvalbumin-expressing (but not calbindin-expressing) GABAergic cells in different brain regions of patients’ postmortem samples [136], reviewed for somatostatin in [137]. Further, postmortem cerebellar and parietal cortices of autistic patients showed a 50% to 60% reduction of the protein expression of GAD65 and GAD67 compared to the postmortem samples obtained from control subjects [138]. Moreover, animal models of epilepsy reported the loss of somatostatin and parvalbumin GABAergic neurons in the hippocampus following kainite- or pilocarpine-induced seizures [139,140]. The hypersynchrony reported in the cortical and hippocampal electroencephalogram (EEG) recordings of hAPPJ20 mouse used as an Alzheimer’s disease mouse model [141] was associated with the reduction of gamma oscillation [142], which depends on the synaptic activity of the fast-spiking parvalbumin-GABAergic neurons [143,144]. As a possible mechanism, the lower expression of the voltage-gated sodium channel subunit Nav1.1 in the parvalbumin-GABAergic neurons was reported in the hAPPJ20 mouse as well as in Alzheimer’s disease patients [142]. Nav1.1 subunit is very critical to the function of parvalbumin-GABAergic neurons as the genetic disruption of such subunits in parvalbumin interneurons increases seizure susceptibility [145]. Interestingly, oxidative stress and/or aging were reported to manipulate the ion channel functions as reviewed in [146]. Consistently, animal models for Alzheimer’s disease as well as aging showed a significant drop of parvalbumin GABAergic neurons in the hippocampus, which was associated with a lower gamma power [147,148]. Further, as described above, the accumulation of lactate and mitochondrial enzyme defects in patients suffering from Huntington’s disease was associated with a selective loss of GABAergic spiny neurons in the striatum compared to the other cholinergic neurons [116]. Despite the fact that Parkinson’s disease is characterized by the loss of dopaminergic neurons of the substantia nigra, the deficit of the pallidal tonic inhibition of some sensory-motor centers, including the thalamus, ventral tegmental area, and substantia nigra, could result in the progression of the non-motor symptoms in Parkinson’s disease [149]. Given that mitochondrial dysfunction was shown in Parkinson’s disease mainly in non-dopaminergic cells [150], this finding could be related to GABAergic dysfunction. Consistent with this idea, depression of GABAergic transmission was shown in a CIQ-induced Parkinson’s disease mouse model [151]. Clinically, the cerebrospinal fluid of Parkinson’s disease patients was characterized by low levels of GABA [152,153].

## 5. The Dilemma of Inhibitory Downregulation in the Aging Central Auditory System

Relatively few studies have examined energy metabolism in the central auditory system during aging. For example, aging-related accumulation of reactive oxygen species or reduction of reactive oxygen species-scavenging enzymes, such as superoxide dismutase, as well as the elevation of mitochondrial DNA mutations, were reported at different regions of the central auditory system, such as the inferior colliculus and auditory cortex [154,155,156]. Such high levels of age-related markers of oxidative stress cause a selective loss of the highly energized GABAergic cells as reported in the inferior colliculus [154]. Therefore, a potential contributor to the aging-related drop of inhibitory transmission could be due to an increase of the vulnerability of GABAergic cells to metabolic stress and/or due to the central plastic changes mediated by peripheral deafferentation (Figure 3). It should also be noted in this regard that some excitatory neurons in the auditory system have extremely high energy demands. For example, excitatory neurons in the superior olive can have firing rates that reach 500 Hz [157,158]. However, these cells appear to be more metabolically resilient than cortical excitatory neurons [159], thus potentially protecting them from aging-related metabolic insults. Interestingly, although tinnitus, which may be related to central disinhibition [160,161], is highly correlated with peripheral hearing loss, it is seen much more frequently with hearing loss in the setting of aging, rather than younger, subjects, suggesting that deafferentation alone is not enough to produce this disorder [162]. In addition, some tinnitus patients have a normal audiogram or hearing with no cochlear synaptopathy [163,164,165] (though a normal audiogram does not rule out all cochlear pathology), which could implicate that the central changes are not necessarily driven by a loss of peripheral acoustic input.

The prognosis of hearing loss and tinnitus is associated with other metabolic disorders, such as diabetes, hyperlipidemia, and hypothyroidism [166,167,168]. It was found that metabolic syndrome negatively affected the recovery of Chinese patients with idiopathic sudden sensorineural hearing loss [166]. Moreover, a long history of hyperlipidemia was associated with the presence of a subtype of tinnitus that is characterized by a high rate of discomfort and inability to habituate [168]. Interestingly, the treatment for hyperlipidemia was found to improve tinnitus [169,170]. Accordingly, the vulnerability of GABAergic cells to metabolic insults could be a significant mediator for their loss or their malfunction during aging. Consistent with this hypothesis, some studies showed that the metabolic perturbation of GABAergic cells could result in a downregulation of their numbers. Animal models of type 2 diabetes showed alteration of the density of GAD67-positive neurons in the striatum and neocortex, and the effects were escalated during aging and alleviated under anti-diabetic treatment [171]. Further, a streptozotocin-induced type 1 diabetic model showed a reduction of glutamic acid decarboxylase-67 (GAD67) and GABAergic receptors in different brain areas [172].

## 6. Future Directions

Some experimental approaches could segregate between the vulnerability of GABAergic cells to metabolic stress and maladaptive plasticity as the main mediator of the downregulation of GABAergic tone in the aging central auditory system. Interestingly, the induction of diabetes to a middle-aged mouse resulted in the disruption of hearing abilities and increasing the amplitude of the inferior colliculus’s response to the broadband noise and tones [173]. As mentioned before, the larger collicular response induced by diabetes could be due to the reduction of central inhibition as a sort of adaptive plasticity due to the peripheral auditory effects of diabetes. However, the decline of inhibition and enhanced excitation in the inferior colliculus observed by Vasilyeva et al. [173] were significant only after four to six months of diabetes induction, which could question if there were ongoing diabetic central changes missed to be observed. Yet, to the best of our knowledge, no study has examined if the pathological metabolic perturbation associated with metabolic syndrome would have a specific negative impact on the performance of GABAergic neurons of the central auditory system. As mentioned earlier, there are many studies correlating the loss of GABAergic cells or their permanent impairment with hypoxia. Accordingly, it would be interesting to monitor the metabolic performance of the GABAergic cells versus non-GABAergic cells in different parts of the central auditory system and correlating their function with hearing performance under some metabolic diseases using animal models for diabetes, ischemia, stroke, or thyroid abnormalities. Monitoring regional GABAergic functions along with the auditory-based behaviors could track the temporal relationship between the GABAergic malfunction in particular brain regions and the onset of hearing loss under those pathological conditions.

Specific lesioning of the GABAergic cells via a conditional deletion of one of their mitochondrial enzymes in different brain regions of the central auditory system could determine the most affected brain area in the central auditory system leading to an auditory perceptual deficit. The selective measurement of the metabolic biomarkers in the GABAergic cells and correlating them with GABAergic cell density, GABA-related biomarkers, network activity, tonotopy as well as the performance of the animals in hearing-related tasks could be ways to examine the impact of the metabolic malfunction of GABAergic cells on the performance of the central auditory system. The metabolic lesioning of GABAergic cells could also help in monitoring the possibility of any mediated plastic changes. Reversing the conditional mutation would provide a good assessment of whether these negative consequences are permanent or could be rescued. A similar approach was used before by deleting the *cox10* gene selectively in parvalbumin neurons (PV-Cox10 CKO), causing a progressive decline in oxidative phosphorylation. This conditional mutation in PV-neurons was associated with impaired sensory gating and sociability without any motor deficit [174].

Although deafening interventions with subsequent measurements of inhibitory synaptic transmission have been done (see above), some modifications of this paradigm would be useful to help disentangle the contributions of hearing loss compared to metabolic factors. For example, it would be instructive to induce hearing loss at several different ages to determine if the likelihood of loss of synaptic inhibition in the setting of hearing loss is also aging-dependent, which would suggest that combinations of factors are at play. In addition, measurement of peripheral markers of stress, such as corticosteroid or catecholamines (or metabolites), or autonomic measurements and determining their relationship to hearing loss would be instructive. Finally, reversible hearing loss interventions, such as those involving earplugs, coupled with longitudinal measurement of synaptic inhibition, as is possible with cranial window/two-photon imaging approaches, could reveal how central inhibition tracks with peripheral hearing loss in real-time. Conducting the previous interventions could distinguish between the vulnerability of GABAergic cells to metabolic insults and plastic changes as two candidates for the downregulation of inhibitory neurotransmission of the central auditory system, and thus provide insights as to the best way to ameliorate pathological central auditory changes that occur with aging.

## Figures and Tables

**Figure 1 brainsci-09-00351-f001:**
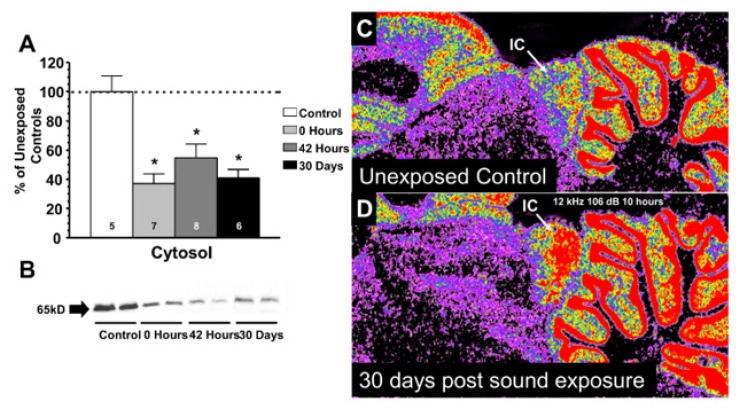
(**A**) Mean glutamic acid decarboxylase (GAD) levels measured via Western blotting of the cytosolic fractions of inferior colliculus homogenates at various times after acoustic trauma. (**B**) Representative autoradiograms from the Western blots taken from the membrane fractions of the inferior colliculus samples at times after acoustic trauma. (**C**) and (**D**) Color-enhanced images of (^3^H) muscimol binding in representative rats without acoustic trauma (**C**) and 30 days after acoustic trauma (**D**), showing the increase in muscimol binding in the inferior colliculus post-trauma. IC = inferior colliculus. Reproduced with permission from [61]. * *p* < 0.05.

**Figure 2 brainsci-09-00351-f002:**
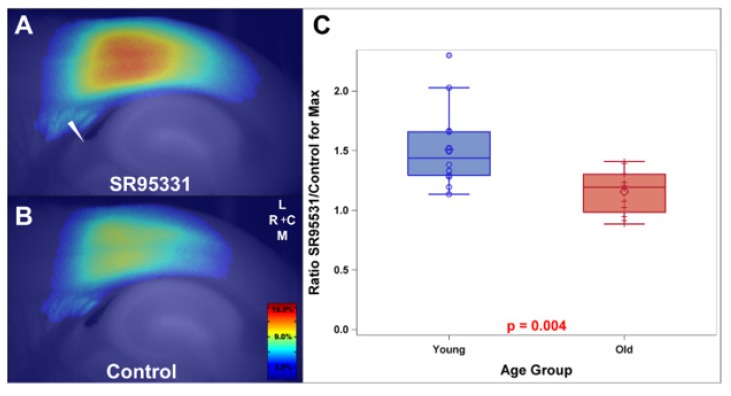
(**A**,**B**) Flavoprotein autofluorescence responses after electrical stimulation of the thalamocortical afferents (triangle) and activation of the auditory cortex under conditions of GABA_A_-receptor blockade (using SR95531, in (**A**) or under control conditions (**B**). Scale bar = 1 mm. (**C**) Ratio of the response to SR9551 to control in young compared to aged mice, indicating a lack of sensitivity to GABA_A_ blockade in the aging auditory cortex. Reproduced with permission from [69].

**Figure 3 brainsci-09-00351-f003:**
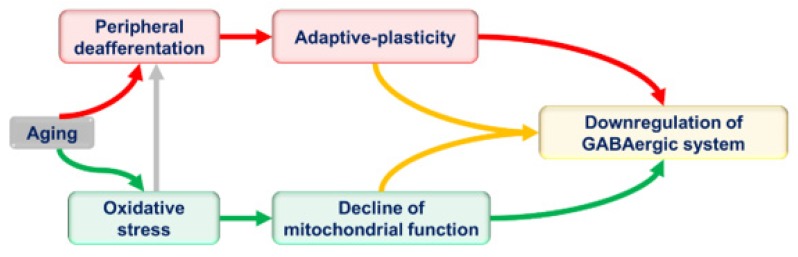
Possible mechanisms for aging-related changes of the GABAergic system. The aging-related downregulation of inhibition of the central auditory system could be mediated by plastic changes (red arrows), metabolic stress and a declining of mitochondrial function (solid green arrows), or a coupling between plastic and metabolic changes (yellow arrows). The aging-related metabolic stress of the cochlea could result in peripheral deafferentation that could be a mechanism to couple metabolic stress with adaptive plasticity (gray arrow).

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
