# Peer review of "Aging and Central Auditory Disinhibition: Is It a Reflection of Homeostatic Downregulation or Metabolic Vulnerability?"

_brainsci, 2019, doi:10.3390/brainsci9120351_

Round 1

Reviewer 1 Report

The review by Ibrahim and Llano articulates a novel hypothesis that aging of the auditory system may also result from age-related disinhibition (loss of GABAergic signalling) that may occur independently from but also contribute to changes that result from loss of auditory input (peripheral deafferentation). This hypothesis draws from recent work linking loss of inhibitory processes with various aging-related behavioural and cognitive deficits. The authors do a great job synthesizing a wide array of evidence for this hypothesis in the central auditory pathway, carefully balancing the two potential mechanisms, comparing findings from across species, and evaluating the consequences. They also clearly outline gaps in knowledge and make suggestions for experiments to fill these gaps. The figures, while simple, are appropriate. I very much enjoyed reading this review and honestly have no suggestions for improvement other than very minor stylistic edits. For example, paragraphs 95-244 and 245-286 are quite long and might benefit from being broken into smaller paragraphs. To conclude, the community of auditory researchers is perhaps too narrowly focused on the role of peripheral deafferentation as the cause of age-related deficits in the auditory system. By carefully constructing arguments for alternative and/or contributing mechanisms, this review makes a very valuable contribution to both auditory and aging sciences by broadening the implications of age-related disinhibition to the auditory system.

Author Response

We thank the reviewers for their careful reviews of the manuscript. We have made all of the suggested changes.

Reviewer 1

The review by Ibrahim and Llano articulates a novel hypothesis that aging of the auditory system may also result from age-related disinhibition (loss of GABAergic signalling) that may occur independently from but also contribute to changes that result from loss of auditory input (peripheral deafferentation). This hypothesis draws from recent work linking loss of inhibitory processes with various aging-related behavioural and cognitive deficits. The authors do a great job synthesizing a wide array of evidence for this hypothesis in the central auditory pathway, carefully balancing the two potential mechanisms, comparing findings from across species, and evaluating the consequences. They also clearly outline gaps in knowledge and make suggestions for experiments to fill these gaps. The figures, while simple, are appropriate. I very much enjoyed reading this review and honestly have no suggestions for improvement other than very minor stylistic edits. For example, paragraphs 95-244 and 245-286 are quite long and might benefit from being broken into smaller paragraphs. To conclude, the community of auditory researchers is perhaps too narrowly focused on the role of peripheral deafferentation as the cause of age-related deficits in the auditory system. By carefully constructing arguments for alternative and/or contributing mechanisms, this review makes a very valuable contribution to both auditory and aging sciences by broadening the implications of age-related disinhibition to the auditory system.

Response: Thank you for the suggestion. Reviewer 2 had a similar suggestion. We now have broken this paragraph up into three paragraphs.

Reviewer 2 Report

General

The manuscript reviews loss of inhibition in the central auditory system that may occur by two different mechanisms: 1) indirect through neuroplasticity after cochlear damage and 2) direct through insults affecting the GABAergic cells in central auditory nuclei. Both mechanisms are assumed to play a role in aging. The focus of the review is on the second mechanism. I like that the two mechanisms are distinguished and discussed, and I like the broader neuroscience context of the auditory topic.

Major comments

1) The review clearly describes how inhibitory neurons are more vulnerable to insults and that insults that are related to neurological diseases or metabolic diseases cause loss of inhibition in auditory nuclei as inferior colliculus or auditory cortex, thus leading to hearing deficits. The link with aging should be described more clearly. How does hearing change during aging while the peripheral organ remains normal (no excessive noise exposures, no ototoxic insults)?

2) The specific effects of central auditory disinhibition on perceptual abilities should be described. For instance, is ability to follow fast modulations (e.g. fast speech) affected?

Minor comments

Line 31. ‘Peripheral amplification’: is meant here ‘hearing aid amplification’?

Line 31-33. The argument does not hold since generally the outer hair cells are affected in cochlear hearing loss, and therefore the frequency tuning is broader than normal, leading to distorted percepts that cannot be countered with amplification. This argument should be removed.

Line 41. ....hemispheric lateralization.. if this is meant.

Line 57-59. The work of Schaette on homeostatic plasticity in the auditory system may be referred to.

Line 74. Work of Suga on sharpening of frequency tuning by lateral inhibition may be referred to.

Line 118. Is this on plugged or unplugged side or on both sides?

Line 134-135. ‘acoustic trauma’ is mentioned twice. Remove one of them.

Line 134-135. What was the effect of acoustic trauma on hearing thresholds? What type of hearing loss are we dealing with?

Line 139. Considering the opposite: is meant ‘increased GABA expression’ instead of ‘lowered GABA expression’?

Line 192. Change ‘data’ to ‘metrics’ to make it more clear.

Line 190-193. Since data of the authors are described, this part may be elaborated.

Page 6. This is a very long paragraph, which makes it harder to read than necessary (content is interesting!). Please split up into 3 or 4 paragraphs.

Line 232-234. A verb seems to be missing in this sentence.

Line 238. Please add reference after ‘reported’.

Line 260. ‘persistent’. Is meant ‘persistence’?

Line 298-299. Are excitatory (non-GABAergic) cells not affected in the cited reports?

Line 315. ‘as described above’: I have not seen something on lactate above, therefore, this may be deleted.

Line 343. Tinnitus and a normal audiogram. Since the frequencies above 8 kHz are clinically not tested, one cannot rule out that tinnitus is caused by loss in the 10-20 kHz region. Further, a 20 dB shift is still regarded within normal range, whereas the underlying cochlear damage may have led to tinnitus. I agree that tinnitus may be caused by purely central mechanisms, however, one must be careful in the arguments.

Line 360. ‘some studies’. Only one study is cited (158). Therefore, please provide some more (or delete the ‘some studies’ phrase.

Line 373-374. ‘the changes shown...’. Please clarify: peripheral changes or central changes?

Author Response

We thank the reviewers for their careful reviews of the manuscript. We have made all of the suggested changes.

Reviewer 2

Major comments

1) The review clearly describes how inhibitory neurons are more vulnerable to insults and that insults that are related to neurological diseases or metabolic diseases cause loss of inhibition in auditory nuclei as inferior colliculus or auditory cortex, thus leading to hearing deficits. The link with aging should be described more clearly. How does hearing change during aging while the peripheral organ remains normal (no excessive noise exposures, no ototoxic insults)?

Response: Thank you for the suggestion. We have expanded the number of areas in the manuscript where we discuss central auditory changes that occur in the absence of hearing threshold changes.

The specific effects of central auditory disinhibition on perceptual abilities should be described. For instance, is ability to follow fast modulations (e.g. fast speech) affected?

Response: Excellent suggestion. We have now included text that describes the functional impact of disinhibition on auditory processing.

Minor comments

Line 31. ‘Peripheral amplification’: is meant here ‘hearing aid amplification’?

Response: Yes, we meant hearing aids but have removed that text based on the next comment.

Line 31-33. The argument does not hold since generally the outer hair cells are affected in cochlear hearing loss, and therefore the frequency tuning is broader than normal, leading to distorted percepts that cannot be countered with amplification. This argument should be removed.

Response: We have removed the relevant lines of text and modified the argument to state that intelligibility is lost out of proportion to threshold and tuning shifts.

Line 41. ....hemispheric lateralization.. if this is meant.

Response: Yes, we meant hemispheric lateralization and have changed the text.

Line 57-59. The work of Schaette on homeostatic plasticity in the auditory system may be referred to.

Response: Thank you for pointing out this work that we neglected. We have now included text references to this work.

Line 74. Work of Suga on sharpening of frequency tuning by lateral inhibition may be referred to.

Response: Good suggestion. We have now included two references to Suga’s work.

Line 118. Is this on plugged or unplugged side or on both sides?

Response: Thank you for catching that omission. On the plugged side. We have modified the text.

Line 134-135. ‘acoustic trauma’ is mentioned twice. Remove one of them.

Response: Done. Thank you.

Line 134-135. What was the effect of acoustic trauma on hearing thresholds? What type of hearing loss are we dealing with?

Response: We have added more detail to this section.

Line 139. Considering the opposite: is meant ‘increased GABA expression’ instead of ‘lowered GABA expression’?

Response: The reviewer is correct. We have changed the text.

Line 192. Change ‘data’ to ‘metrics’ to make it more clear.

Response: Thank you. We respectfully changed “data” to “findings.” Using “metrics” here did not seem appropriate, since its use would imply that we are referring the testing method rather than the results.

Line 190-193. Since data of the authors are described, this part may be elaborated.

Response: We have added more text to this section.

Page 6. This is a very long paragraph, which makes it harder to read than necessary (content is interesting!). Please split up into 3 or 4 paragraphs.

Response: Good suggestion. Done.

Line 232-234. A verb seems to be missing in this sentence.

Response: Thank you. We have re-written the sentence

Line 238. Please add reference after ‘reported’.

Response: Done

Line 260. ‘persistent’. Is meant ‘persistence’?

Response: Correct. We have changed this to “persistence”

Line 298-299. Are excitatory (non-GABAergic) cells not affected in the cited reports?

Response: The cited report in this case did not report on changes in non-GABA cells in general, but did report on calbindin-positive cells, which did not change. We have modified the text to reflect this finding.

Line 315. ‘as described above’: I have not seen something on lactate above, therefore, this may be deleted.

Response: Done

Line 343. Tinnitus and a normal audiogram. Since the frequencies above 8 kHz are clinically not tested, one cannot rule out that tinnitus is caused by loss in the 10-20 kHz region. Further, a 20 dB shift is still regarded within normal range, whereas the underlying cochlear damage may have led to tinnitus. I agree that tinnitus may be caused by purely central mechanisms, however, one must be careful in the arguments.

Response: We agree and have now added some qualifying language to indicate that a normal audiogram does not rule out all cochlear pathology.

Line 360. ‘some studies’. Only one study is cited (158). Therefore, please provide some more (or delete the ‘some studies’ phrase.

Response: Thank you. We had omitted the citation in the subsequent sentence. We have now included it.

Line 373-374. ‘the changes shown...’. Please clarify: peripheral changes or central changes?

Response: We have clarified that this sentence is referring to central changes.